# Synthesis and Properties of Magnetic Aryl-Imidazolium Ionic Liquids with Dual Brønsted/Lewis Acidity

**DOI:** 10.3390/ma11122539

**Published:** 2018-12-13

**Authors:** Jui-Cheng Chang, Che-Hsuan Yang, I-Wen Sun, Wen-Yueh Ho, Tzi-Yi Wu

**Affiliations:** 1Department of Chemical and Materials Engineering, National Yunlin University of Science and Technology, Yunlin 64002, Taiwan; wuty@yuntech.edu.tw; 2Bachelor Program in Interdisciplinary Studies, National Yunlin University of Science and Technology, Yunlin 64002, Taiwan; 3Department of Chemistry, National Cheng Kung University, Tainan 70101, Taiwan; yangvul3rm03@gmail.com (C.-H.Y.); iwsun@mail.ncku.edu.tw (I.-W.S.); 4Department of Cosmetic Science, Chia Nan University of Pharmacy & Science, Tainan 71710, Taiwan

**Keywords:** tunable aryl-imidazolium Brønsted/Lewis magnetic ionic liquids, magnetic susceptibility, magnetic coupling constant, hysteresis, Brønsted/Lewis properties

## Abstract

A series of unique tunable aryl-imidazolium magnetic ionic liquids (MILs) with dual acidity that contain both Brønsted and Lewis acidic sites (abbreviated as B-L MILs) were synthesized and characterized using nuclear magnetic resonance and mass spectrometry. Physical properties, such as thermal properties, magnetic susceptibility, and Brønsted and Lewis acidity, were measured. These properties were found to depend on the cation structure. These B-L MILs had good solubility in many organic solvents, good thermal stability, and low melting points, and exhibited magnet-like behavior. For these B-L MILs, the Brønsted acidity was measured using ultraviolet-visible (UV-Vis), and the Lewis acidity was measured using Fourier transform infrared spectroscopy (FTIR). The results showed that B-L MILs with an electron-withdrawing group in the aryl-imidazolium moiety had higher Brønsted acidity, whereas those with an electron-donating group had higher Lewis acidity. This type of ionic liquid, with both Brønsted and Lewis acidic sites, is expected to be a useful solvent and catalyst for organic reactions.

## 1. Introduction

Ionic liquids (ILs) have been shown to be very promising green solvents with several general advantages such as extremely low volatility, non-flammability, low melting points, high thermal, chemical, and electrochemical stability, and extraordinary solvent properties compared to those of traditional organic solvents. ILs are typically composed of large asymmetric organic cations and various inorganic or organic anions; therefore, the physical-chemical properties of ILs can be tuned by changing the types of cations and anions [1,2,3,4]. ILs are widely applied in organic synthesis [5,6], advanced energy technology [7,8], environmental chemistry [9,10], medicine [11,12], extraction [13,14], and nanotechnology [15,16].

In addition to common ILs, many interesting ILs with special properties have been reported, such as dicationic ILs [17,18,19,20,21], polymeric ILs [13,14,22,23], magnetic ILs [24,25,26,27], and acidic ILs [28,29,30]. Studies have established a relationship between the structure and properties of ILs. Among studied ILs, imidazolium salts are the most prominent class of ILs, usually carrying *sp*^3^-hybridized carbon atoms as substituents at both nitrogen atoms of the heterocycle [31,32]. These *sp*^3^-hybridized carbon atoms are strongly related to the tuning of the physical-chemical properties of ILs and are involved in the van der Waals interactions. In 2009, the first tunable aryl-alkyl ionic liquids (TAAILs) were reported by Strassner et al. (shown in Scheme 1) [33,34]. These TAAILs have a combination of *sp*^3^ alkyl and *sp*^2^ aryl substituents at the nitrogen atoms of the imidazolium core [33,34,35]. The physical-chemical properties of these ILs can be tuned via the specific electronic effects that result from the σ- and π-system even substituent effect in the aryl moiety. This means that the TAAILs are not restricted to van der Waals interactions; they also allow π-π interactions [34]. A large number of studies on the properties and applications of TAAILs have thus been published [36,37].

Magnetic ILs (MILs) are room-temperature ILs formed from organic cations and inorganic metal anions with unpaired electrons [38]. MILs have inherent paramagnetic properties (i.e., there is no need to add magnetic particles). The magnetic properties of 1-ethyl-3-methylimidazolium tetrachloroferrate ([C_2_mim][FeCl_4_]) were initially emphasized by Yoshida and Saito [39]. The temperature-dependent paramagnetic properties of 1-butyl-3-methylimidazolium tetrachloroferrate ([C_4_mim][FeCl_4_]) were investigated by Hamaguchi et al. [40]. MILs are primarily based on high-spin Fe(III) *d*^5^ in the form of tetrachloroferrate(III) with various counter cations. MILs can be adsorbed on a magnet and have a certain magnetization in the presence of an external magnetic field owing to their high single-ion magnetic moments. In 2008, the relationship between the structure and magnetic behavior of transition-metal-based ILs was reported by Sesto et al. [41]. It was found that different cationic structures with the same anion ([FeCl_4_]^−^) have different properties, that is, the magnetic susceptibilities of [PR_4_][FeCl_4_] (R = decyl) and [C_10_mim][FeCl_4_] are different. [C_10_mim][FeCl_4_] does not respond strongly to a Nd_2_Fe_14_B magnet, whereas [PR_4_][FeCl_4_] droplets do and remain intact almost indefinitely (in a water environment). MILs are mostly used for extraction [27] and separation [31]. For example, 1-butyric acid-3-methylimidazolium tetrachloroferrate(III) [C_3_H_6_COOHmim][FeCl_4_] was used as both an extraction solvent and a catalyst for the oxidative extraction of benzothiophene from an oil sample. Following the extraction process, this MIL was separated from the oil sample by the application of a magnetic field and subsequently distilled to remove the extracted components [27]. In another example, [Bmim][FeCl_4_] was used as supported MIL membranes (SMILMs) for gas separation; the influence of the application of an external magnetic field on CO_2_/N_2_ and CO_2_/air permeability through SMILMs in the range of 0–1.5 T was evaluated [31]. MILs with [FeCl_4_]^−^ anion can serve as an effective solvent, catalyst, and/or template in organic syntheses, such as aryl Grignard cross-coupling (under mild and air-stable conditions; they are safe and have high recyclability) and the one-pot synthesis of the medicinal heterocyclic compound 1-amidoalkyl-2-naphthol (they shorten the reaction time and give a high yield) [42,43,44].

ILs with acidic properties (called AILs) are an important branch of ILs [45]. The acidity is generally of the Brønsted or Lewis type [29,45]. The acidic functions or groups can be in the cation, anion, or both. AILs, which are non-volatile and non-corrosive, are mostly used as catalysts in organic syntheses to overcome the disadvantages (e.g., corrosion, contamination, and recycling difficulty) of traditional strong inorganic acids such as H_2_SO_4_, H_3_PO_4_, and V_2_O_5_ [46,47]. For example, Lewis AILs such as [CholineCl][ZnCl_2_]_3_ has been used as in a Friedel–Crafts acylation [48], allowing chemo- and region-selective acylation of aromatic compounds and five-membered heterocycles under mild conditions [48]. In addition, Brønsted AILs such as [MIM-PS]HSO_4_, where MIM-PS is 1-(4-sulfonic acid) propyl-3-methylimidazolium, was used for the esterification of n-butanol with acetic acid [49], giving a high conversion efficiency along with the increase of reaction temperature and time. Because Brønsted AILs have one or more acidic hydrogen(s) residing on N or O atoms, they are applied as catalysts for organic reactions related to the activation of carbonyl groups, such as esterification and Michael addition [46]. Meantime, Lewis AILs have electron-accepting ability for metals (such as Fe^3+^, Al^3+^) in an anion group, and therefore, they are used for organic derivatization reactions of aromatic compounds such as Friedel–Crafts alkylation and acylation [46].

AILs with a combination of Brønsted and Lewis acidic characteristics in the same molecule are known as dual-functionalized AILs [29,50,51]. Dual Brønsted/Lewis ILs have been used to simultaneously carry out catalysis and extraction in the oxidative desulfurization of diesel fuel. For example, the AIL [Hnmp]Cl/ZnCl_2_ (Hnmp = H-*N*-methylpyrrolidonium) was synthesized and applied in a desulfurization reaction by Yu et al. [50]. This AIL can be used for catalysis and extraction and has high sulfur removal efficiency and high recyclability. AILs have also been used in the synthesis of [HO_3_S–(CH_2_)_3_–NEt_3_]Cl–FeCl_3_ for the one-pot synthesis of biodiesel from waste oil [51]. Brønsted/Lewis AILs with tunable acidity are becoming increasingly important.

The present study synthesizes a series of tetrachloroferrate salts, in which each tetrahalometallate(III) anion is associated with a tunable aryl-imidazolium (no alkyl group) cation synthesized from imidazole and iodobenzene, and then protonated by hydrochloride acid. These tunable aryl-imidazolium magnetic ILs simultaneously have Brønsted and Lewis acidic sites and have the unique characteristics of one acidic hydrogen residing on N atoms and electron-accepting ability for Fe^3+^ in the anion group (these ILs are denoted hereafter as B-L MILs). Since these B-L MILs have a structure with tunable aryl-imidazolium moiety, they are expected to exhibit physical properties different from the imidazolium cation carrying the alkyl group as substituents at both nitrogen atoms. In this study, the substituent effects are also introduced into tunable aryl-imidazolium moiety. The thermal properties, solubility in organic solvents, magnetic behavior, and Brønsted and Lewis acidity of these B-L MILs are discussed.

## 2. Experiments

### 2.1. Chemicals

Imidazole (purity: 99%), iodobenzene (purity: 98%), 1-iodo-4-methylbenzene (purity: 98%), 1-iodo-4-methoxybenzene (purity: 98%), 1-iodo-4-nitrobenzene (purity: 98%), cesium carbonate (Cs_2_CO_3_, purity: 99%), and iron(III) chloride (FeCl_3_, purity: 99%) were purchased from Alfa Aesar (Ward Hill, MA, USA). Copper acetate (Cu(OAc)_2_·H_2_O, purity: 99%) was purchased from Showa (Saitama, Japan). Hydrochloride acid (HCl, purity: 37%) was purchased from Scharlab (Sentmenat, Barcelona, Spain). *p*-Xylene was purchased from Sigma-Aldrich (St. Louis, MO, USA). Benzyl chloride was purchased from Acros (Morris Plains, NJ, USA). Dimethyl sulfoxide (DMSO), acetonitrile (ACN), hexane (Hex), ethyl acetate (EA), methanol (MeOH), ethanol (EtOH), dichloromethane (DCM), and ethyl ether (Et_2_O) were purchased from J. T. Baker (Edmonton, Alberta, Canada) (all ACS grade, ≤0.01% H_2_O). All chemicals were purchased from a commercial supplier and used as received.

### 2.2. General Procedure for Synthesis of Target FeCl_4_ Anion Salts (5a–5d)

Imidazole (1.0 mmol) was mixed with iodobenzene (2a, 1.2 mmol) or its derivatives (1-iodo-4-methylbenzene for 2b, 1-iodo-4-methoxybenzene for 2c, and 1-iodo-4-nitrobenzene for 2d), copper(II) acetate (0.1 mmol), and cesium carbonate (2.0 mmol). All chemicals were dissolved in 20 mL DMSO and stirred under nitrogen gas at 110 °C for one day. Then, the reaction was stopped and the solution was cooled to room temperature. This completed reaction was extracted three times with ethyl acetate and deionized water. The water layer was also repeated to extract three times by ethyl acetate and all organic layers were collected. The organic layers were rotatory concentrated and their precipitate was purified using column chromatography with ethyl acetate/hexane (*v*/*v* = 1/3) to yield 3a (white solid). The procedure for obtaining 3b–3d (all white solids) was identical to that used for 3a. Compound 3a (1.0 mmol) was mixed with hydrochloride acid (2.0 mmol), dissolved with ethanol, and stirred in an ice bath for two hours. The solvent was evaporated and then the residue was washed by ethyl ether to yield compound 4a. The same procedure was used to obtain 4b–4d. All compounds (4a–4d) were white solids. Compounds 4a–4d (all 1.0 mmol), respectively, were reacted with iron(III) chloride (1.0 mmol) and stirred in ethanol at room temperature for five hours. The pressure was reduced to evaporate the solvent, ethyl acetate was added, and then the solution was centrifuged for 10 min. The supernatant was collected to the concentrate solvent under vacuum to obtain yellow solids (5a–5d).

1-phenyl-1H-imidazole (3a) (PhIm): Yield 88%. ^1^H NMR (500 MHz, CDCl_3_) δ = 7.19 (s, 1H), 7.27 (s, 1H), 7.35–7.48 (m, 5H), 7.84 (s, 1H). ^13^C NMR (125 MHz, CDCl_3_) δ = 118.2, 121.5, 127.5, 129.8, 130.3, 135.5, 137.3.

1-(4-methyphenyl)-1H-imidazole (3b) (PhImCH_3_): Yield 89%. ^1^H NMR (500 MHz, CDCl_3_) δ = 2.40 (s, 3H), 7.19 (s, 1H), 7.24–7.27 (m, 5H), 7.81 (s, 1H). ^13^C NMR (125 MHz, CDCl_3_) δ = 20.9, 118.3, 121.4, 130.2, 130.3, 135.0, 135.6, 137.4.

1-(4-methoxyphenyl)-1H-imidazole (3c) (PhImOCH_3_): Yield 88%. ^1^H NMR (500 MHz, CDCl_3_) δ = 3.85 (s, 3H), 6.98 (d, *J* = 8.8 Hz, 2H), 7.19 (d, *J* = 8.8 Hz, 2H), 7.30 (d, *J* = 9.2 Hz, 2H), 7.76 (s, 1H). ^13^C NMR (125 MHz, CDCl_3_) δ = 55.6, 114.9, 118.7, 123.2, 130.0, 130.7, 135.8, 158.9.

1-(4-nitrophenyl)-1H-imidazole (3d) (PhImNO_2_): Yield 86%. ^1^H NMR (500 MHz, CDCl_3_) δ = 7.29 (s, 1H), 7.38 (s, 1H), 7.59 (d, *J* = 8.8 Hz, 2H), 7.98 (s, 1H), 8.39 (d, *J* = 9.2 Hz, 2H). ^13^C NMR (125 MHz, CDCl_3_) δ = 117.6, 121.0, 125.7, 131.7, 135.4, 142.0, 146.3.

3-phenyl-1H-imidazolium chloride (4a) [HPhIm]Cl: Yield: 95%. ^1^H NMR (500 MHz, CDCl_3_) δ = 7.50–7.66 (m, 7H), 9.67 (s, 1H). ^13^C NMR (125 MHz, CDCl_3_) δ = 119.9, 121.0, 122.3, 130.4, 130.6, 134.0, 134.7.

3-methylphenyl-1H-imidazolium chloride (4b) [HPhImCH_3_]Cl: Yield: 96%. ^1^H NMR (500 MHz, CDCl_3_) δ = 2.44 (s, 3H), 7.38 (d, *J* = 8.4 Hz, 2H), 7.45–7.48 (m, 3H), 7.60 (s, 1H), 9.42 (s, 1H). ^13^C NMR (125 MHz, CDCl_3_) δ = 21.1, 120.0, 121.0, 122.1, 131.1, 132.2, 133.6, 140.9.

3-methoxyphenyl-1H-imidazolium chloride (4c) [HPhImOCH_3_]Cl: Yield: 95%. ^1^H NMR (500 MHz, CDCl_3_) δ = 3.89 (s, 3H), 7.08 (d, *J* = 8.8 Hz, 2H), 7.40 (s, 1H), 7.49 (d, *J* = 7.6 Hz, 2H), 7.56 (s, 1H), 9.17 (s, 1H). ^13^C NMR (125 MHz, CDCl_3_) δ = 55.8, 115.6, 120.4, 121.1, 123.9, 127.6, 133.4, 161.0.

3-nitrophenyl-1H-imidazolium chloride (4d) [HPhImNO_2_]Cl: Yield: 94%. ^1^H NMR (500 MHz, D_2_O) δ = 7.60 (s, 1H), 7.83 (d, *J* = 9.2 Hz, 2H), 7.91 (s, 1H), 8.40 (d, *J* = 8.8 Hz, 2H), 9.23 (s, 1H). ^13^C NMR (125 MHz, D_2_O) δ = 120.9, 121.1, 123.4, 125.7, 134.4, 139.6, 148.0.

3-phenyl-1H-imidazolium tetrachloroferrate(III) (5a) [HPhIm][FeCl_4_]: Yield: 99%. Analytical calculation for C_9_H_9_Cl_4_FeN_2_: C, 31.53; H, 2.65; N, 8.17. Found: C, 31.38; H, 2.75; N, 8.16%. LRMS-ESI (+ve): *m*/*z* [M]^+^ calcd for C_9_H_9_N_2_^+^: 145.08; found 145.08. LRMS-ESI (−ve): *m*/*z* [M]^−^ calcd for [FeCl_4_]^−^: 197.81; found: 198.01.

3-methylphenyl-1H-imidazolium tetrachloroferrate(III) (5b) [HPhImCH_3_][FeCl_4_]: Yield: 99%. Analytical calculation for C_10_H_11_Cl_4_FeN_2_: C, 33.66; H, 3.11; N, 7.85. Found: C, 33.90; H, 3.22; N, 7.60%. LRMS-ESI (+ve): *m*/*z* [M]^+^ calcd for C_10_H_11_N_2_^+^: 159.09; found 159.08. LRMS-ESI (−ve): *m*/*z* [M]^−^ calcd for [FeCl_4_]^−^: 197.81; found: 197.81.

3-methoxyphenyl-1H-imidazolium tetrachloroferrate(III) (5c) [HPhImOCH_3_][FeCl_4_]: Yield: 99%. Analytical calculation for C_10_H_11_Cl_4_FeN_2_O: C, 32.21; H, 2.97; N, 7.51. Found: C, 31.91; H, 3.00; N, 7.50%. LRMS-ESI (+ve): *m*/*z* [M]^+^ calcd for C_10_H_11_N_2_O^+^: 175.09; found 175.08. LRMS-ESI (−ve): *m*/*z* [M]^−^ calcd for [FeCl_4_]^−^: 197.81; found: 197.81.

3-nitrophenyl-1H-imidazolium tetrachloroferrate(III) (5d) [HPhImNO_2_][FeCl_4_]: Yield: 99%. Analytical calculation for C_9_H_8_Cl_4_FeN_3_O_2_: C, 27.87; H, 2.08; N, 10.83. Found: C, 27.88; H, 2.37; N, 10.95%. LRMS-ESI (+ve): *m*/*z* [M]^+^ calcd for C_9_H_8_N_3_O_2_^+^: 190.06; found 190.05. LRMS-ESI (−ve): *m*/*z* [M]^−^ calcd for [FeCl_4_]^−^: 197.81; found: 197.81.

### 2.3. Characterization

The ^1^H spectra of the purified cross-coupling products (for 3a–3d and 4a–4d) were recorded in CDCl_3_ (for 3a–3d and 4a–4c) and D_2_O (for 4d) (Cambridge Isotope Laboratories Inc., Tewksbury, MA, USA, 99.9% D) on a nuclear magnetic resonance (NMR, model: Bruker AVANCE-500, Hamburg, Germany) spectrometer at 500 MHz at a temperature of 25 °C (NMR spectra for 4a–4d are shown in Appendix A). The structures of the synthesized tunable aryl-imidazolium magnetic ILs with a Brønsted/Lewis moiety were identified directly using an elemental analyzer (model: Vario EL III, Langenselbold, Germany), mass spectrometry (cation and anion moieties, quadrupole time-of-flight mass spectrometer, model: Xevo G2 QTOF, Milford, MA, USA), ultraviolet-visible (UV-Vis) spectroscopy (anion moiety, solvent: CH_3_CN, UV-Vis, model: JASCO V-630, Tokyo, Japan), and Raman spectroscopy (anion moiety, model: Thermo Scientific DXR^TM^, Erlangen, Germany; diode laser excitation beam: 780 nm). Since these B-L MILs have a paramagnetic Fe^3+^ ion, they cannot be characterized using NMR. Thermal decomposition temperatures were determined using thermogravimetry analysis (TGA, model: SDT Q600, New Castle, DE, USA; used gas: 1 atm N_2_ gas) with a heating rate of 5 °C/min (the *T*_d_ selected at a given temperature correspond to *T*_d_ onset, 5%, and 10% weight loss). Melting points were measured using differential scanning calorimetry (DSC, model: Shimadzu DSC-60, Kyoto, Japan; used gas: 1 atm N_2_ gas) and a melting point apparatus (model: Yanagimoto MP-S3, Kyoto, Japan) with heating rates of 5 °C/min and 1 °C/min, respectively. Variable-temperature direct-current magnetic susceptibility and hysteresis loop measurements were performed for the B-L MILs, which were restrained in eicosane to prevent torque, on a superconducting quantum interference device vibrating sample magnetometer (SQUID VSM, model: Quantum Design MPMS-7, San Diego, CA, USA) with a 7-T magnet, operated in the range of 2–300 K.

### 2.4. Acidity Measurement

The measurement of Brønsted acidity was conducted using UV-Vis spectroscopy with a basic indicator following procedures described in previous studies [52,53,54]. The crystal violet indicator (pKa = 0.8) and each of the B-L MILs (5a–5d) were dissolved with deionized water at concentrations of 0.0123 mM and 5 mM, respectively. The Brønsted acidity strength of each of the B-L MIL solutions is expressed by the Hammett acidity function (*H_o_*) [52,53,54]. The *H_o_* value was calculated as follows:*H_o_* = pK (I) + log ([I]/[IH^+^])(1)
where pK (I) is the pKa value of the indicator prepared to an aqueous solution, and [I] and [IH^+^] are the molar concentrations of the unprotonated and protonated forms of the indicator in the solvent, respectively [52,53].

The Lewis acidity of the B-L MILs was determined using Fourier transform infrared spectroscopy (FTIR, model: Perkin Elmer Spectrum RX1, Waltham, MA, USA) [55]. IR spectra were collected for mixtures of B-L MILs and acetonitrile with a ratio of 200:600 (200 mg B-L MIL:600 μL CH_3_CN, stirred 5 h in a closed sample bottle before IR measurement, two CaF_2_ salt tablets and one drop sample were sandwiched to measure). A new transmittance peak appeared at around 2310 cm^−1^ in the IR spectra of B-L MIL + acetonitrile mixtures in addition to that of pure acetonitrile, which indicates the coordination of acetonitrile to the Lewis acidic center. The intensity of this peak was used to characterize the Lewis acidity of B-L MILs.

## 3. Results and Discussion

### 3.1. Synthesis and Characterization of B-L MILs

The synthesis procedure and the product yields are summarized in Scheme 2. The chemical structures are shown in Table 1. B-L MILs 5a–5d were synthesized in three steps. First, an Ullmann-type [56] coupling reaction was used to combine imidazole and iodobenzene moieties (2a–2d, in the presence of Cs_2_CO_3_, 10 mol % Cu(OAc)_2_, and in DMSO solvent at 110 °C) to give 3a–3d with high isolated yields (88–96%). In the second step, 3a–3d (aryl imidazole) were dissolved in ethanol and treated with hydrochloric acid in an ice bath to yield Brønsted acidic ionic liquids 4a–4d. The chloride ions in 4a–4d were coordinated with FeCl_3_·6H_2_O in ethanol in the final step to obtain 5a–5d with good yields. Of note, the second and final steps in this synthesis route show an atom economy of close to 100%.

The target samples, 5a–5d, were directly characterized using elemental analysis, electrospray ionization mass spectrometry (ESI-MS), UV-Vis spectroscopy, and Raman absorption spectroscopy. The presence of the [FeCl_4_]^−^ anion was confirmed by the mass spectrum (Figure 1), UV-Vis spectrum (Appendix A), and Raman spectrum (Appendix A). The negative ion mode of ESI-MS for 5a shown in Figure 1a exhibits a fragment peak at *m*/*z* = 198.01, corresponding to [FeCl_4_]^−^ (calcd. for 197.81). Figure 1b shows a fragment peak at *m*/*z* = 145.08 ([HPhIm]^+^ cation) in positive ion mode for 5a (mass spectra of B-L MILs 5b–5d are shown in Appendix A). The UV-Vis absorption spectrum shown in Appendix A exhibits three characteristic bands of the [FeCl_4_]^−^ anion, at 534 (t_1u_ to t_2g_* at FeCl_4_ orbital [57]), 619 (t_1g_ to e_g_* [57]), and 688 nm (t_1g_ to t_2g_* [57]), respectively, similar to those reported by Hamaguchi [40,58]. The Raman spectrum in Appendix A exhibits a strong band at approximately 330 cm^−1^, which can be assigned to the totally symmetric stretching mode (A1 symmetry vibration by DFT calculation) of [FeCl_4_]^−^ [40,58,59,60]. The frequencies have very slight difference between these four B-L MILs (5a: 330 cm^−1^, 5b: 332 cm^−1^, 5c: 332 cm^−1^, 5d: 333 cm^−1^) and this may be attributed to the fact that the [FeCl_4_]^−^ anion has interacted with the aryl-imidazolium cation with different para-substitutions. These ESI-MS and spectroscopy results prove that the synthesized ILs contained both Brønsted (proton in cation) and Lewis ([FeCl_4_]^−^ anion) acidic sites. Elemental analysis results showed that all B-L MILs, 5a–5d, contained C, H, and N in the expected ratios. The B-L MILs (for example, 5a) also exhibited a strong response to a magnet. As shown in Appendix A, 5a formed a layer under the ethyl ether layer in a glass sample tube. This 5a layer was strongly attracted to a NdFeB (0.55-T) magnet; its shape became highly distorted, becoming triangular (see Appendix A).

### 3.2. Solubility of B-L MILs

In order to evaluate the applicability of B-L MILs for catalysis, their solubility in several solvents was determined; the results are summarized in Table 2. In this study, “soluble” means that 5 mg of the B-L MIL can be completely dissolved in 1 mL of the solvent at room temperature. All B-L MILs exhibit good solubility in the solvents tested, except hexane. Even in DCM and ethyl ether, B-L MILs show slight solubility. The [FeCl_4_]^−^-anion-containing ILs synthesized in this study easily dissolved in deionized water in a few seconds. This solubility is higher from that reported in previous studies; for example, [C_10_mim][FeCl_4_] required several hours to dissolve in deionized water [41]. This may be attributed to the easy dissociation of the proton bonded to the N atom of aryl-imidazolium in water. It also implies hydrogen bonding between B-L MILs (hydrogen bonded to N atoms and C2-imidazolium) and solvents (oxygen or nitrogen atoms). It was found that in a solvent with a high dielectric constant, B-L MILs dissolve easily, but in a solvent with a dielectric constant of lower than 9 (except EA), B-L MILs are only slightly soluble.

### 3.3. Thermal Stability of B-L MILs

The TGA curves of B-L MILs (5a–5d) and their corresponding chloride precursors (4a–4d) are shown in Figure 2. As shown in this Figure and in Table 1, the B-L MILs are thermally more stable than their precursors because the precursors contain chloride anions, which exhibit high nucleophilicity at high temperature. B-L MILs with a sp^2^-substituted phenyl ring at the nitrogen atom have a specific charge distribution that results in good thermal stability [34,35,61,62]. From Table 1 and Figure 2, the first thermal decomposition step involves the chloride ion of the [FeCl_4_]^−^ anion eliminating two protons from the aryl-imidazolium cation (one proton was bonded to the N atom and the other was in the C2-imidazolium core) [63,64]. This behavior arises in B-L MILs 5c and 5d. The TGA results indicate that the B-L MILs with the resonance effect in para-substitution, such as that with the –OCH_3_ and –NO_2_ groups (5c and 5d), in the aryl moiety of the cation both have more obviously second decomposition behavior, ranging from 300 °C to 600 °C. 

### 3.4. Melting Point Properties of B-L MILs

All B-L MILs synthesized in this study were yellow solids at room temperature. Figure 3 shows the DSC curves for B-L MILs. As shown in Table 3, the melting points obtained using DSC and a capillary melting point apparatus are in similar ranges. The order of melting points is 5a > 5d ~ 5c > 5b. The electron-withdrawing group in para-substitution (e.g., –NO_2_ group, 5d) has a higher melting point than that of the electron-donating group (–CH_3_ group, 5b). However, the melting point of the electron-donating –OCH_3_ group (5c) is unexpectedly close to that of 5d. The para-substitution effects on the melting points of TAAILs have been previously discussed [65] based on para-substitution computational calculations (B3LYP) of dipole moments of TAAILs, which are related to the melting points. The results of dipole moments indicated different polarizations of the imidazolium cores and the electronic effect of the substituent [34,65,66]. They also indicated that para-substitution is not restricted to the phenyl ring; it also influences the conjugated π-system of the imidazolium core. Studies have shown that an electron-donating substituent has a lower melting point than that of an electron-withdrawing substituent because of the high dipole moment in the latter, such as that in para-nitro-substitution. An unexpectedly high melting point in the presence of the –OCH_3_ group has been reported, but still lower than that with –NO_2_ (the ESP (electrostatic surface potentials) calculation shows that the dipole moment of the –OCH_3_ group in para-substitution is lower than that of the –NO_2_ group) [65]. The melting points observed in the present study may be more consistent with the literature.

### 3.5. Magnetic Properties of B-L MILs

The temperature dependence of the DC (constant magnetic field) magnetic susceptibilities of powder samples of B-L MILs 5a–5d measured in the temperature range of 2.0–300 K under an applied field of 1.0 kOe is shown in Figure 4. For 5a, the value steadily decreases from 4.85 emu K mol^−1^ at 300 K to 0.34 emu K mol^−1^ at 2 K. 5b–5d show similar temperature dependence trends. The χ*_M_*T (magnetic susceptibility) values estimated from Figure 4 are shown in Table 4.

The spin-only effective magnetic susceptibility value, μ*_eff_*, can be defined in terms of the *g* value (anisotropy) and the *S* value (total electron spin) as follows [67,68,69]:*g*[*S*(*S* + 1)]^1/2^ = μ*_eff_* = 2.828(χ*_M_*T)^1/2^(2)

The experimental μ*_eff_* values (shown in Table 4) of 5a, 5b, and 5d are higher than the theoretical spin-only values (5.92 for the [FeCl_4_]^−^ anion) (*g* = 2.0) with a non-interacting metal center, with *S* = 5/2 for [FeCl_4_]^−^. The value for 5c is slightly less than 5.92 because its electron-donating (–OCH_3_) group with the resonance effect results in increased opportunity for more electrons of the cationic moiety to interact with the [FeCl_4_]^−^ anion. The magnetism of the [FeCl_4_]^−^ anion is well described by the Curie–Weiss Equation [67]:χ*_M_*^−1^ = (*T* − Θ)/*C*(3)
where *C* is the Curie constant and Θ is the Weiss constant. The Θ value can be expressed in terms of the magnetic coupling constant (*J*) [67] as follows:Θ = *zJS*(*S* + 1)/3*k*(4)

The Weiss constant values obtained from the fitting (shown in Figure 5) for B-L MILs 5a–5d are shown in Table 4. The negative values indicate that these salts have antiferromagnetic interaction. 

The data in Table 4 also show that the B-L MIL without a substituent in the aryl-imidazolium moiety (5a) has a more negative Θ value than those of the B-L MILs with substituents (5b–5d). The Θ value of 5c, with the resonance effect of the –OCH_3_ group, is smaller than that of 5b. The para-substitution effects have been previously discussed based on ESP calculations, which are related to the electron density. It also indicates that the increase of the electron density on the aryl-imidazolium moiety due to the –OCH_3_ group is more than that due to the –CH_3_ group [65]. This result is consistent with the order of spin-only effective magnetic susceptibility values. Of note, 5d, with the resonance effect of the electron-withdrawing (–NO_2_) group, has a high μ*_eff_* value but the unexpectedly smallest Θ value among the B-L MILs. All B-L MILs synthesized in this work except 5c have magnetic susceptibility values that are higher than those previously reported (references in Table 4) for salts with the [FeCl_4_]^−^ anion. Some studies have reported that the Θ value is somewhat related to the nearest neighbor M–Cl**^…^**Cl–M distance (the antiferromagnetic pathway through superexchange interactions) in an anion with a single metal; that is, a shorter M–Cl**^…^**Cl–M distance in general results in a larger Θ value [69,70].

The B-L MILs exhibit magnet-like behavior in their magnetic hysteresis curves. The isothermal magnetizations *M* (H) of these B-L MILs were measured at 300 K up to 20 kOe (shown in Appendix A). The magnetization seems to show a linear dependence on the applied magnetic field. In this figure, 5c has the lowest magnetization among the B-L MILs. The fitted lines of the B-L MILs actually have magnet-like behavior. Appendix A shows a small hysteresis loop at 300 K with a coercive field (Hc, 5a: 301 Oe, 5b: 239 Oe, 5c: 402 Oe, and 5d: 325 Oe). This magnet-like behavior seems to be an important breakthrough in this field [24,39,40,58]. A more detailed understanding of the magnetic behavior of the B-L MILs requires knowledge of their crystal structures. Unfortunately, the good solubility of these ILs in many organic solvents (see Table 2) makes it difficult to obtain good B-L MIL crystals for study. Various methods are currently being tried to obtain crystals.

### 3.6. Brønsted Acidity Properties of B-L MILs

The B-L MILs have protons bonded to the N atom and C2-imidazolium in the aryl-imidazolium moiety; therefore, the Brønsted acidity properties were evaluated based on the Hammett acidity function (*H_o_*) using Equation (1). UV-Vis spectroscopy with crystal violet as the indicator was used to determine the [I]/[HI^+^] ratio from the measured absorbance difference of the indicator before and after the addition of B-L MILs. The representative UV-Vis spectra of *H_o_* values are shown in Figure 6. The maximal absorbance of the unprotonated form of crystal violet in deionized water is at 591 nm. The *H_o_* values for the B-L MILs obtained using Equation (1) are listed in Table 5. As shown, the B-L MIL without a substituent in the aryl-imidazolium moiety has the lowest acidity (5a, highest *H_o_* value) and that with an electron-withdrawing substituent has the highest acidity (5d, lowest *H_o_* value). The electron-withdrawing group –NO_2_ in B-L MIL 5d may allow the proton bonded to the N atom in the aryl-imidazolium moiety to leave easily. B-L MIL 5c also has a high *H_o_* value because the electron-donating group –OCH_3_ increases the electron density on the aryl-imidazolium moiety, making it more difficult to release the proton. Of note, 5b, with the electron-donating group –CH_3_, also shows high acidity, although its acidity is slightly lower than that of 5d. This may be due to the inductive effect in the –CH_3_ group, which leads to a lower electron density of the aryl-imidazolium moiety compared to that with the –OCH_3_ group [65]. Table 5 shows that the B-L MILs exhibit low *H_o_* values which may be due to the para-substitution effects. Therefore, the B-L MILs synthesized in this study are expected to be useful catalysts for organic reactions such as esterification.

### 3.7. Lewis Acidity Properties of B-L MILs

The B-L MILs have the [FeCl_4_]^−^ complex anion that contains a Lewis acid, Fe^3+^. However, the Lewis acidity could be affected by the cations of the IL. The Lewis acidity of the B-L MILs was measured using FTIR. Figure 7 shows the FTIR spectra of B-L MILs + acetonitrile mixtures. A previous study indicated that acetonitrile added to [BMIM]Cl/FeCl_3_ results in the appearance of a new band at 2310 cm^−1^ [55]. In this study, a new absorption peak also appears in only para-substitution of electron-donating groups at approximately 2310 cm^−1^ in comparison with the absorptions at 2252 and 2292 cm^−1^ of pure acetonitrile in the FTIR spectra of these mixtures compared to that of pure acetonitrile, indicating the coordination of acetonitrile with Fe^3+^. The intensity of this peak is related to the Lewis acidity of B-L MILs [55]. Based on the intensity of this peak, B-L MILs that have an electron-donating group (–OCH_3_ in 5b and –CH_3_ in 5c) were found to have high acidity. This confirms that tuning the cationic structures of B-L MILs can affect their acidity. Due to electrostatic interaction, the proton of the para-nitro-substitution (5d) system can easily leave the cation, inhibiting the formation of hydrogen bonds between cations and anions; therefore, Fe^3+^ does not have an empty orbital to accept the lone pair electron from CH_3_CN (shown in Scheme 3a). In contrast, this proton in B-L MILs 5b and 5c rarely leaves and [FeCl_4_]^−^ can form a hydrogen bond with it (shown in Scheme 3b). The proposed effect of para-substitution (Scheme 3) corresponds to the results of Raman shift even if there are very slight differences of Raman shift between these four B-L MILs (5a: 330 cm^−1^, 5b: 332 cm^−1^, 5c: 332 cm^−1^, 5d: 333 cm^−1^). The empty orbital of Fe^3+^ in the anion can make CH_3_CN coordinate easily, resulting in the high Lewis acidity of 5b and 5c.

## 4. Conclusions

[FeCl_4_]^−^ anion ILs based on a tunable aryl-imidazolium moiety with dual Brønsted and Lewis acidic sites (B-L MILs) were synthesized. These B-L MILs exhibited good solubility in many organic solvents and deionized water, but not in hexane. All B-L MILs had better thermal stability than that of their precursors, which contained the chloride anion. In general, the B-L MILs showed two thermal decomposition steps. The second thermal decomposition step seemed to be related to the substituent in the aryl-imidazolium moiety. The substituents with the resonance effect (–OCH_3_ and –NO_2_) in para-substitution had more obviously second decomposition behavior (300–600 °C). The melting points of all B-L MILs were below 100 °C. The order of melting points could be attributed to the electronic effect of the substituent. Regarding the magnetic properties, the B-L MIL that contained an electron-donating substituent with the resonance effect had the lowest magnetic susceptibility and magnetization. The striking magnetic-like behavior is that these B-L MILs had a small hysteresis loop at 300 K. Acidity measurements indicated that the B-L MIL without a substituent and those with an electron-withdrawing substituent (–NO_2_) in the aryl-imidazolium moiety had high Brønsted acidity. B-L MILs with electron-donating substituents (–CH_3_ and –OCH_3_) had high Lewis acidity. Because the B-L MILs have good solubility in many organic solvents, it is difficult to find a proper solvent to crystallize them. Therefore, the crystal structure of these ILs was not provided here, but it will be reported when available. These ILs, which contain both Brønsted and Lewis acid sites, are expected to have potential applications in catalysis and organic syntheses, such as esterification, oxidative desulfurization, and Friedel–Crafts alkylation.

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
