# Peer review of "Synthesis and Properties of Magnetic Aryl-Imidazolium Ionic Liquids with Dual Brønsted/Lewis Acidity"

_materials, 2018, doi:10.3390/ma11122539_

Round 1
Reviewer 1 Report
In the present manuscript, the authors synthesized magnetic ionic liquids, potentially useful for catalysis. The paper is well written and meets the standards of publication for the journal. Hence, I recommend its publication after including the suggestions detailed below:
Introduction section
1. In general, for the introduction section: It lacks a lot of references supporting the statements made in this section. I highlighted some of them in the points below, but I recommend the authors to go through the introduction and make sure all the statements are supported by references.
2. The properties assigned to the ILs in the first paragraph are generalization. Not all the ILs meet such properties, since there a million of combinations that can result in ionic liquids. I’d suggest the authors to be more accurate when describing ILs.
3. When describing the properties of ILs and the possibility to tune their properties, please include references for the readers to go to seminal papers of ILs.
4. Second paragraph: The term “traditional ILs” is confusing. If the authors are referring to a specific terminology proposed previously, please include the references. If not, please modify such statement.
5. Page 3: The sentence “A large number of studies on the properties and applications of TAAILs have thus been published.” Should be supported by references.
6. The sentence “Magnetic ILs (MILs) are room-temperature ILs formed from organic cations and inorganic metal anions with unpaired electrons. Should be followed by references for seminal papers about magnetic ionic liquids.
7. Page 4: The “greenness” of ILs should be seriously considered, and statements such as “MILs, especially those with an anionic Fe3+ complex (usually the [FeCl4]- anion), can serve as a green, inexpensive, and reaction-safety solvent, (…)” aren’t accurate. These properties can be tuned and will be based not only on the cation but also on the cation (specially greenness, cost, toxicity, etc.). Please modify this sentence.
Experimental section
8. Include affiliation of the different vendors/suppliers
9. Section 2.2 should be synthesis of the MIL
10. Page 11: Characterization of 3-phenyl-1H-imidazolium tetrachloroferrate (III) (5a) [HPhIm][FeCl4]: Why was the found mass in the LRMS-ESI (-ve) higher than the calculated one? If compared with the other masses determined, this difference of 0.2 between calculated and determined seems to be significant.
Results:
11. Include after each table footnotes indicating the meaning of abbreviations.
12. Section: Brønsted acidity properties of B-L MILs – For comparison, it would be interesting to also determine the Bronsted acidity properties of the chloride-based synthesized components (4a-4d) and evaluate the effect of the anion on this property.
13. Section 3.7: For comparison, it would be interesting to also determine the Lewis acidity properties of the chloride-based synthesized components (4a-4d) and demonstrate the increase of this property with [FeCl4]-.
14. Section 3.7: The spectrum from the compound 5a should be discussed.
Conclusions: How would the magnetic properties be affected if those ILs are used for catalytic reactions, since it is the iron the one that has magnetic and catalytic properties at the same time.
Author Response
Responses to reviewers’ comments
Dear Prof. Eduardo, Editor Ariel Zhou, and reviewers
Thank you very much for the useful comments and suggestions from you and the reviewers on our manuscript. We have modified the manuscript accordingly; changes are marked in yellow in the revised manuscript (called Materials-400990 original R1). And the responses to the reviewers’ comments are listed below point by point:
Sincerely yours,
Jui-Cheng Chang. Assistant Professor
Department of Chemical and Materials Engineering,
and Bachelor Program in Interdisciplinary Studies,
National Yunlin University of Science and Technology,
Yunlin 64002, Taiwan, R.O.C.
TEL: +886-5-5342601 ext: 4685
Reviewer 1' comments:
Introduction section
1. In general, for the introduction section: It lacks a lot of references supporting the statements made in this section. I highlighted some of them in the points below, but I recommend the authors to go through the introduction and make sure all the statements are supported by references.
Response:
Thank you very much for this constructive comment. We have added new references to supplement the informations point by point as below
2. The properties assigned to the ILs in the first paragraph are generalization. Not all the ILs meet such properties, since there a million of combinations that can result in ionic liquids. I’d suggest the authors to be more accurate when describing ILs.
Response:
Thank you very much for this incisive comment. We have added new references in this version as described below:
(1) Organic synthesis (refs.[5, 6]):
[5] Ionic Liquids as Tool to Improve Enzymatic Organic Synthesis. Chem. Rev. 2017, 117, 10567-10607.
[6].Brønsted acidic ionic liquids: Green catalysts for essential organic reactions. Journal of Molecular Liquids 218 (2016) 95-105.
(2) advanced energy technology (refs. [7, 8]).
[7]. Electric double layer capacitors of high volumetric energy based on ionic liquids and hierarchical-pore carbon. J. Mater. Chem. A, 2014, 2, 14963-14972.
[8]. An ether bridge between cations to extend the applicability of ionic liquids in electric double layer capacitors. J. Mater. Chem. A, 2016, 4, 19160-19169.”
(3) environmental chemistry (refs. [9, 10]).
[9]. Tuning environmental friendly chelate-based ionic liquids for highly efficient and reversible SO2 chemisorption. ACS Sustainable Chem. Eng., 2018, 6 (11), pp 15292-15300.
[10]. Supported Absorption of CO2 by Tetrabutylphosphonium Amino Acid Ionic Liquids. Chem. Eur. J. 2006, 12, 4021-4026.
(4) medicine (refs. [11, 12]).
[11]. Biological Activity of Ionic Liquids and Their Application in Pharmaceutics and Medicine. Chem. Rev. 2017, 117, 7132-7189.
[12]. Ionic liquids for energy, materials, and medicine. Chem. Commun., 2014, 50, 9228-9250.”
(5) extraction (refs. [13, 14]).
[13]. Conductive polymeric ionic liquids for electroanalysis and solid-phase microextraction. Analytica Chimica Acta 910 (2016) 45-52.
[14]. Electropolymerized Pyrrole-Based Conductive Polymeric Ionic Liquids and their Application for Solid-Phase Microextraction. ACS Appl. Mater. Interfaces, 2017, 9 (29), 24955-24963.”
(6) nanotechnology (refs. [15, 16]).
[15]. Dual Phase-Controlled Synthesis of Uniform Lanthanide-Doped NaGdF4 Upconversion Nanocrystals Via an OA/Ionic Liquid Two-Phase System for In Vivo Dual-Modality Imaging. Adv. Funct. Mater. 2011, 21, 4470-4477.
[16]. Oxygen reduction electrocatalysts sophisticated by using Pt nanoparticle-dispersed ionic liquids with electropolymerizable additives. J. Mater. Chem. A, 2018, 6, 11853-11862.”
3. When describing the properties of ILs and the possibility to tune their properties, please include references for the readers to go to seminal papers of ILs.
Response:
Thanks for this comment. We have added one reference (ref. [1]) “Wasserscheid, P.; Welton, T. Ionic Liquids in Synthesis; Wiley-VCH: Weinheim, 2008.”
4. Second paragraph: The term “traditional ILs” is confusing. If the authors are referring to a specific terminology proposed previously, please include the references. If not, please modify such statement.
Response:
Thanks for this comment. The term “traditional ILs” has been changed to “common ILs”.
5. Page 3: The sentence “A large number of studies on the properties and applications of TAAILs have thus been published.” Should be supported by references.
Response:
Thanks for this comment. New references:
[36]. Tunable aryl alkyl ionic liquids (TAAILs) based on 1-aryl-3,5-dimethyl-1H-pyrazoles. Journal of Molecular Liquids 248 (2017) 314-321.
[37]. Cobalt-Catalyzed Hydroarylations and Hydroaminations of Alkenes in Tunable Aryl Alkyl Ionic Liquids. Org. Lett. 2018, 20, 6215-6219.” have been added.
6. The sentence “Magnetic ILs (MILs) are room-temperature ILs formed from organic cations and inorganic metal anions with unpaired electrons. Should be followed by references for seminal papers about magnetic ionic liquids.
Response:
Thanks for this comment. A new reference [38] has been added.
[38]. Molecular modeling and physicochemical properties of 1-alkyl-3-methylimidazolium-FeX4 and -Fe2X7 (X = Cl and Br) magnetic ionic liquids. Journal of Molecular Liquids 256 (2018) 175-182.
This added reference is marked as yellow.
7. Page 4: The “greenness” of ILs should be seriously considered, and statements such as “MILs, especially those with an anionic Fe3+ complex (usually the [FeCl4]- anion), can serve as a green, inexpensive, and reaction-safety solvent, (…)” aren’t accurate. These properties can be tuned and will be based not only on the cation but also on the cation (specially greenness, cost, toxicity, etc.). Please modify this sentence.
Response:
Thanks for this comment. The word ““green” has been deleted and the statement has been modified as “MILs with [FeCl4]- anion can serve as an effective solvent, catalyst, and/or template in organic syntheses”.
A new reference [44] has also been added in this version:
[44]. Lewis Acidic Ionic Liquid [Bmim]FeCl4 as a High Efficient Catalyst for Methanolysis of Poly (lactic acid). Catal Lett (2017) 147:2298-2305.
Experimental section
8. Include affiliation of the different vendors/suppliers
Response:
Thanks for this comment. We have added the vendors or suppliers in every label of chemical as below:
Alfa Aesar (Ward Hill, Massachusetts), Showa (Saitama, Japan), Scharlau (Sentmenat, Barcelona), Aldrich (St. Louis, MO), and Acros (Morris Plains, NJ)
9. Section 2.2 should be synthesis of the MIL
Response:
Thanks for this comment. We have re-organized the experimental section and the section 2.2 is changed as “General procedure for synthesis of target FeCl4- anion salts (5a-5d)” in page 6-10.
10. Page 11: Characterization of 3-phenyl-1H-imidazolium tetrachloroferrate (III) (5a) [HPhIm][FeCl4]: Why was the found mass in the LRMS-ESI (-ve) higher than the calculated one? If compared with the other masses determined, this difference of 0.2 between calculated and determined seems to be significant.
Response:
Thanks for observing this difference. Actually, this reason is unclear currently and we are indeed temporarily unable to provide detail information for this difference between theoretical and calculated values of mass spectrum.
We can only assume that this difference of 0.2 is within the error (< 5%) currently and confirms the anion is [FeCl4]- from mass spectrum.
Results:
11. Include after each table footnotes indicating the meaning of abbreviations.
Response:
Thanks reviewer 1 very much for providing this good comment. We have added the footnotes in every Table as below:
Table 1: (Td: thermal decomposition temperature)
Table 2: (full names of organic solvents are shown in section 2.1 Chemicals)
Table 3: (DSC: differential scanning calorimetry, MP-S3: melting point apparatus)
Table 4: (χMT: magnetic susceptibility, μeff : spin-only effective magnetic susceptibility, Θ: Weiss constant)
Table 5: (Amax: the maximum absorption, [I]: the concentration of indicator, [IH+]: the concentration of protonated indicator, Ho: Brønsted acidity)
12. Section: Brønsted acidity properties of B-L MILs - For comparison, it would be interesting to also determine the Bronsted acidity properties of the chloride-based synthesized components (4a-4d) and evaluate the effect of the anion on this property.
Response:
Thanks reviewer 1 very much for this comment. The Bronsted acidity properties of the chloride-based synthesized components (4a-4d) will be investigated and compared to ILs 5a-5d in our future works on their applications in organic catalysis.
13. Section 3.7: For comparison, it would be interesting to also determine the Lewis acidity properties of the chloride-based synthesized components (4a-4d) and demonstrate the increase of this property with [FeCl4]-.
Response:
Thanks for this comment. In general, ILs containing only halide anions are considered Lewis basic. The Lewis acidcity of the chloride-based components (4a-4d) will be investigated and compared to ILs 5a-5d in our future works on their applications in organic catalysis.
14. Section 3.7: The spectrum from the compound 5a should be discussed.
Response:
(1) The para-subsititution of 5a is only H-atom. H atom is only spherical orbital (s orbital) and this orbital has the characteristic of electron attraction (like “electron-withdrawing property”). Therefore, The Lewis acidity behavior of 5a is similar with 5d.
(2) The compound 5a itself has lower Lewis acidity (or low acid amount to form Lewis acid) to result in the lowest new peak (2310cm-1) that we are unable to find.
The above-mentioned two possible points might be proved by our future study on the use of the ILs in organic catalysis.
Conclusions: How would the magnetic properties be affected if those ILs are used for catalytic reactions, since it is the iron the one that has magnetic and catalytic properties at the same time.
Our response:
We thank reviewer 1 very much for this good issue. We will discuss this point in our future work in application of these ILs for organic catalysis.

Reviewer 2 Report
The manuscript by Chang et al (materials-400990) focuses on the synthesis and characterization of Aryl-Imidazolium Ionic Liquids with magnetic properties. Overall, the project is well described. This manuscript can be accepted in present status after following minor corrections/suggestions.
1. Introduction: In general, applications of polymeric ILs is an emerging field. It is recommended to discuss and include literature in the introduction. Ex: ACS Applied Materials & Interfaces 9 (29), 24955-24963 and Analytica Chimica Acta 910, 45-52.
2. Authors concluded the potential applications of these ILs in catalysis and organic syntheses, such as esterification, oxidative desulfurization, and Friedel-Crafts alkylation. It is better if authors can expand this idea by giving more explanation in the discussion with a general example.
Author Response
Responses to reviewers’ comments
Dear Prof. Eduardo, Editor Ariel Zhou, and reviewers
Thank you very much for the useful comments and suggestions from you and the reviewers on our manuscript. We have modified the manuscript accordingly; changes are marked in yellow in the revised manuscript (called Materials-400990 original R1). And the responses to the reviewers’ comments are listed below point by point:
Sincerely yours,
Jui-Cheng Chang. Assistant Professor
Department of Chemical and Materials Engineering,
and Bachelor Program in Interdisciplinary Studies,
National Yunlin University of Science and Technology,
Yunlin 64002, Taiwan, R.O.C.
TEL: +886-5-5342601 ext: 4685
Reviewer 2' comments:
1. Introduction: In general, applications of polymeric ILs is an emerging field. It is recommended to discuss and include literature in the introduction. Ex: ACS Applied Materials & Interfaces 9 (29), 24955-24963 and Analytica Chimica Acta 910, 45-52.
Response:
Thanks reviewer 2 very much for this comment. We have added these two references (re-numbered [13, 14]) as well as another two references ([22] and [23] as listed below) in second paragraph of introduction, line 2.
22. Clark, K.D.; Emaus, M.N.; Varona, M.; Bowers, A.N.; Anderson, J.L. Ionic liquids: solvents and sorbents in sample preparation. J. Sep. Sci. 2018, 41, 209-235.
23. Clark, K.D.; Anderson, J.L. Ionic liquids as tunable materials in (bio) analytical chemistry. Anal. Bioanal. Chem. 2018, 410, 4565-4566.
2. Authors concluded the potential applications of these ILs in catalysis and organic syntheses, such as esterification, oxidative desulfurization, and Friedel-Crafts alkylation. It is better if authors can expand this idea by giving more explanation in the discussion with a general example.
Response:
We thank reviewer 2 very much for this comment. We have added new description relating to the applications of these ILs in catalysis and organic syntheses, such as esterification, oxidative desulfurization, and Friedel-Crafts alkylation. “For example, Lewis AILs such as [CholineCl][ZnCl2]3 has been used as in a Friedel-Crafts acylation [48], allowing chemo- and region-selective acylation of aromatic compounds and five-membered heterocycles under mild conditions [48]. In addition, Brønsted AILs such as [MIM-PS]HSO4, where MIM-PS is 1-(4-sulfonic acid) propyl-3-methylimidazolium, has been used for the esterification of n-butanol with acetic acid [49], giving a high conversion efficiency along with the increase of reaction temperature and time.” (in page 5, marked as yellow)

Reviewer 3 Report
Dear Editor
The manuscript with the ID materials-400990 deals with the synthesis and characterization of a series of magnetic room temperature ionic liquids with both Bronsted and Lewis acidity. In particular, the authors prepare four RTILs with different substituent on aryl moiety and they investigate the relationships among the substituent nature and thermal stability, melting points, magnetic properties and acidic properties.
The work appears interesting, and for this reason I recommend the publication without revision.
Just a few comments on the text:
1) table 1 should be more compact excluding the formulas
2) pag. 16 line 7: probably is figure 2
3) page 18 third last line: probably is figure 4
Author Response
Responses to reviewers’ comments
Dear Prof. Eduardo, Editor Ariel Zhou, and reviewers
Thank you very much for the useful comments and suggestions from you and the reviewers on our manuscript. We have modified the manuscript accordingly; changes are marked in yellow in the revised manuscript (called manuscript-revised). And the responses to the reviewers’ comments are listed below point by point:
Sincerely yours,
Jui-Cheng Chang. Assistant Professor
Department of Chemical and Materials Engineering,
and Bachelor Program in Interdisciplinary Studies,
National Yunlin University of Science and Technology,
Yunlin 64002, Taiwan, R.O.C.
TEL: +886-5-5342601 ext: 4685
Reviewer 3' comments:
1. Table 1 should be more compact excluding the formulas
Our response:
Thanks to reviewer 3 for this comment. Formulas have been removed from Table 1.
2. Page. 16 line 7: probably is Figure 2
Our response:
Thanks. The error has been corrected.
3. Page 18 third last line: probably is Figure 4
Response:
Thanks. The error has been corrected.
